# Impact of feralization on evolutionary trajectories in the genomes of feral cat populations

**María Esther Nieto-Blázquez[1]***, **Manuela Gómez-Suárez[1]**, **Markus Pfenninger** [1], **Katrin Koch[2]***

**1** Department of Molecular Ecology, Senckenberg Biodiversity and Climate Research Centre, Frankfurt am Main, Germany, **2** Department of Biodiversity, Conservation and Attractions, Former, Biodiversity and Conservation Science, Woodvale, Australia

* menietoblazquez@gmail.com (MEN-B); koch@kafmano.de (KK)

**Data Availability Statement:** All relevant data are within the manuscript and its Supporting information files.

## Abstract

Feralization is the process of domesticated animals returning to the wild and it is considered the counterpart of domestication. Molecular genetic changes are well documented in domesticated organisms but understudied in feral populations. In this study, the genetic differentiation between domestic and feral cats was inferred by analysing whole-genome sequencing data of two geographically distant feral cat island populations, Dirk Hartog Island (Australia) and Kaho'olawe (Hawaii) as well as domestic cats and European wildcats. The study investigated population structure, genetic differentiation, genetic diversity, highly differentiated genes, and recombination rates. Genetic structure analyses linked both feral cat populations to North American domestic and European cat populations. Recombination rates in feral cats were lower than in domestic cats but higher than in wildcats. For Australian and Hawaiian feral cats, 105 and 94 highly differentiated genes compared to domestic cats respectively, were identified. Annotated genes had similar functions, with almost 30% of the divergent genes related to nervous system development in both feral groups. Twenty mutually highly differentiated genes were found in both feral populations. Evolution of highly differentiated genes was likely driven by specific demographic histories, the relaxation of the selective pressures associated with domestication, and adaptation to novel environments to a minor extent. Random drift was the prevailing force driving highly divergent regions, with relaxed selection in feral populations also playing a significant role in differentiation from domestic cats. The study demonstrates that feralization is an independent process that brings feral cats on a unique evolutionary trajectory.

## Introduction

Feralization is the understudied counterpart of domestication, and refers to the return of domestic animals to natural habitats [1, 2]. The process often involves a fundamental increase in both natural and sexual selection with strong effects on several components of fitness in

**Funding:** The research was partially funded by the Landes-Offensive zur Entwicklung Wissenschaftlich ökonomischer Exzellenz (LOEWE) Program of the Hessian Ministry of Higher Education, Research, Science and the Arts through the LOEWE Centre for Translational Biodiversity Genomics (LOEWE-TBG). The funders had no role in study design, data collection and analysis, decision to publish, or preparation of the manuscript.

**Competing interests:** The authors have declared that no competing interests exist.

order to thrive in a novel ecological environment [2–5]. Although feralization has previously been described as the reversal of domestication [5, 6], the process rarely results in the complete return to an "ancestral" form [6, 7]. The reappearance of pre-domestication traits has been shown not to necessarily rely on the same genetic variation present in undomesticated (ancestral) populations, but rather the result of other genetic mechanisms under novel selection pressures [2, 7].

Multiple populations of domestic animals of a wide taxonomic range have undergone this process. In many cases, the feralization process has led to invasiveness of these feral species which highly negatively impact global biodiversity [2, 8, 9]. A proper understanding of feralization presents numerous challenges, for example the inference of the times in which feralization took place, a phenomenon that probably followed domestication from the earliest moments and continued up to the present day.

Expectations on molecular genetic changes are well documented in organisms that have undergone domestication but remain elusive for feral taxa. Specifically, the domestication process often results in reduced genetic diversity as selective breeding imposes successive genetic bottlenecks and artificial selection on target phenotype(s) [10]. Moreover, recombination rates have been found to increase among domestics as compared to their wild counterparts that is also driven by strong artificial selection [11]. Although there is growing evidence for further bottlenecking among populations becoming feral (i.e. dingoes; [12]) due to selective sweeps (i.e. chickens; [6], how the release of domestic animals in the wild impacts recombination rates remains largely unexplored. Single nucleotide polymorphisms (SNPs), the most common type of genetic variation, have emerged as a powerful tool for investigating processes such as feralization [13]. SNP markers offer several advantages in the study of feralization because they are abundant throughout the genome, allowing the assessment of genome-wide patterns of genetic diversity within and between feral and domestic populations [14]. Additionally, SNPs can be used to identify loci that have undergone selection during feralization by comparing allele frequencies between populations [15]. Furthermore, SNP data can be harnessed to reconstruct the evolutionary history of feral populations, providing insights into their source populations and dispersal patterns [16]. By analyzing SNP variation, researchers can gain a deeper understanding of the genetic mechanisms underlying adaptation in feral species, ultimately contributing to a broader knowledge of the complex dynamics of feralization.

One of the most widespread and detrimental invasive predators are feral cats (*Felis catus*) [8, 17–19]. Cats became globally distributed after their initial domestication from *Felis lybica* in the Near East and Egypt in association with early human settlements and the arising need to control rodent pests [17, 20, 21]. Eventually, the extensive dispersal of cats was possible through deliberate and stowaway travel on naval vessels [22]. As a consequence, the current domestic cat's global distribution [8, 17–19] is the result of centuries of explorers, sealers, whalers and colonist bringing them on voyages for controlling rodents and/or as companions, which reached even the remotest islands since 1800 [19, 23]. Given their relatively human-independent life-style, the establishment of feral cat populations in various regions of the world was probably unavoidable.

Feral cats are a highly successful invasive species, thriving in diverse environmental conditions and successfully preying on small vertebrates and even larger insects [24–27], with extensive negative impacts on local ecosystems and in certain places such as Australia the impact of cats is recognized as one of the most important fauna conservation issues [24]. For example, the largest island off the Western Australian coast, Dirk Hartog Island, has lost 10 of 13 native terrestrial mammal species presumably through predation by cats [28]. The domestication history of cats and their multiple introductions globally within the last 200 years provide an

excellent opportunity to observe "natural replicates" of the genomic response to feralization from a known and largely identical domesticated starting point [17, 21, 29–31].

Here, we aim to study the patterns of genomic differentiation between domestic and feral cats, and investigate genome-wide patterns of the independent feralization process in two geographically distant islands not inhabited by humans. The absence of humans, and therefore further sources of domestic cats, makes these two island ideal systems to study feralization in isolation in the absence of gene flow with domestic cats. In particular we are asking 1) what is the origin of the two feral cat populations and how differentiated are they from their ancestral populations? 2) How did feralization impact molecular patterns and processes in comparison to domesticated cats and their original wild ancestors? And 3) which evolutionary forces shaped the divergence given the recent introduction (ca. 200 years) of domestic cats in the study areas and did these forces target the same genes and/or traits in both feral populations?

## Materials and methods

### Sampling, data processing and variant calling

**Sampling.**   We included four feral cat samples per island from Dirk Hartog Island (Australia) and Kaho'olawe (Hawaii) (S1 Table in S1 File). Domestic cats were introduced in these islands in the last 200 years [32] and currently have very limited to no human inhabitants. Although Dirk Hartog Island is used for ecotourism, visitor numbers are very low and a genetic study on the feral cat population showed no indication for introduction of cats to the island in the near past [28]. Both islands have a semi-arid climate [32–34] with an annual rainfall of 220mm (Dirk Hartog Island) and 600mm (Kaho'olawe) per year. Mean maximum daily temperatures range from 38.8 ˚C in summer and 21.8 ˚C during winter on Dirk Hartog Island, and 26.3 ˚C with no distinct differences between seasons on Kaho'olawe (Bureau of Meteorology, Australia). Vegetation on both islands is generally sparse, low and open [32–34] indicating similar environmental regimes. Despite these similarities, Dirk Hartog is ca. three times larger than Kaho'olawe (Fig 1a). Trapping and sampling details of feral cats in Dirk Hartog Island can be found in [28]. In addition, we included 8 German wildcat individuals (*Felis silvestris*) in this study. We obtained fastq files for the eight feral and eight wildcats from the ENA (European Nucleotide Archive) project PRJEB40421 [35]. In addition, we obtained 11 fastq files for 11 non-fancy short hair breed domestic cat individuals (S1 Table in S1 File) from the ENA project PRJNA343389 (data as part of the 99 Lives Cat Genome Sequencing Initiative, 2016). In total, whole genome sequences of 27 individuals were included in the present study.

**Filtering and SNP calling.**   Raw reads of all 27 samples were quality checked using FastQC. No trimming of reads was necessary. Reads were then mapped to the latest *F. catus* reference genome version Felis_catus_9.0 (GenBank assembly accession GCA_000181335.4) using BWQ mem v.0.7.15 [36] and results checked using QualiMap v.2.2.1 [37]. We used Platypus v.1.0 [38] to call SNPs (single nucleotide polymorphisms). Bases with quality scores below 30 and reads with mapping quality below 30 were ignored, and only variants with at least 6 reads were kept. We filtered for biallelic variants passing all filters and pruned all SNPs for Linkage disequilibrium (LD) with a squared correlation coefficient of more than 0.5 using BCFtools v.1.9 [39] in 100-kb windows.

### Population structure, population differentiation and phylogenetic reconstruction analyses

We performed a principal component analysis (PCA) using the unlinked SNPs on the complete dataset (feral, domestic and wildcat individuals) using the R package *factoextra* v.1.0.7. In

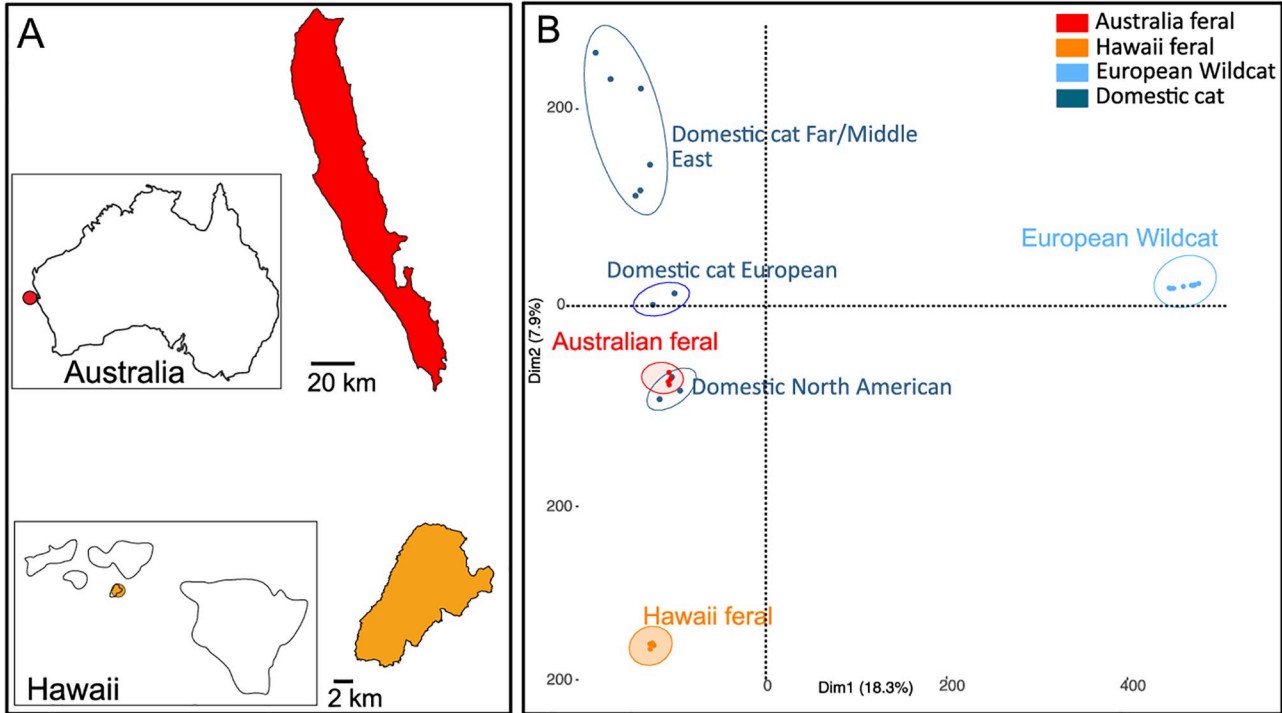

**Fig 1.** a) Map of sampling locations in Australia and Hawaii with representation of the islands, b) Principal component analyses (PCA) based on genome-wide unlinked SNPs complete dataset. Coloration indicating the origin of the individual (Australian feral cats = red, Hawaiian feral cats = orange, European wildcat = light blue and Domestic cat = dark blue).

addition, we used ADMIXTURE v. 1.3.0 [40], a maximum likelihood estimation approach to infer ancestry proportions using an increasing number of clusters ($K$), from $K = 1$ to $K = 5$, withdefault settings and a 10-fold cross validation (CV) procedure. We used the unlinked SNPs on a reduced dataset which included feral cat groups and European and North American domestic cats and plotted results from both with PONG v.1.5 [41].

We estimated the genome-wide fixation index ($F_{ST}$) [42] for each feral cat group and domestic cats using non-overlapping 10-kb windows with VCFtools v.0.1.17 [43]. We retained $F_{ST}$ above 99.8% CI (confidence interval) for the functional selection analyses. Manhattan plots were created using R package *qqman* [44]. Additionally, we calculated the absolute divergence ($D_{XY}$) using non-overlapping 100-kb windows with *popgenWindows.py* (https://github.com/simonhmartin/genomics_general/blob/master/popgenWindows.py). To determine the phylogenetic relationships amongst groups and individuals, we used a Maximum Likelihood (ML) approach with RAxML v.8.2.12 [45], nucleotide substitution model GTR+ Γ and a rapid bootstrap (BS) algorithm with 1000 replicates. We used the filtered SNP-dataset and re-rooted the tree using the Near East domestic cats of the dataset [46].

## Genetic diversity

We estimated nucleotide diversity (π) per cat group (Hawaiian feral, Australian feral, domestic cat and wildcat) with VCFtools v.0.1.17 [43] using 100-kb windows. We also estimated Watterson's theta ($\theta_W$) per cat group with angsd v.0.931 [47] using 100-kb windows. We assessed pairwise differences among $\theta_W$ between groups with a double-sided Bayesian paired t-test

implemented in R package *BayesianFirstAid* [48]. In addition, we estimated Tajima's *D* ($T_D$) per cat group with VCFtools v.0.1.17 using 100-kb windows.

## Recombination rate analyses

We compared recombination rates amongst Hawaiian feral cats, Australian feral cats, wildcats and a random set of domestic cats. We used ReLERNN [49], a recurrent neural networks (RNN)-based method for estimating the genomic map of recombination rates directly per chromosome from a genotype alignment for the three groups independently. We used the SNP dataset with biallelic variants and default settings. First, ReLERNN_SIMULATE split the. vcf file by chromosome and ran the simulations using coalescent simulation program *msprime*, to simulate training, validation, and test data sets. We used a window size of 100-kb on the unphased SNP dataset. Then ReLERNN_TRAIN used the simulations created to train a recurrent neural network; we used 250 number of epochs to train over and 10 validation steps. ReLERNN_PREDICT predicts per-base recombination rates in non-overlapping windows across all chromosomes of the genome. We finally obtained 95% CI around each predicted recombination rate with ReLERNN_BSCORRECT. The pairwise differences among recombination rates between the different groups were assessed again with *BayesianFirstAid* [48].

## Runs of Homozygosity (ROH) analyses

We identified ROHs in order to get a better idea on the inbreeding levels in feral cat populations relative to domestic source, but also wildcats. We first removed duplicates, and readgroups were added using Picard v.2.20.8 (https://broadinstitute.github.io/picard/). Genotypes were called using BCFtools v.1.10.2 *mpileup* and *call* [39]. For this, sites with a mapping and a base quality higher than 20 were kept. Afterwards, BCFtools v.1.10.2 *filter* was used to keep sites with a minimum read depth of 3, and to filter for indels and for sites with more than 5% missing data. The final combined data set consists of 2,427,089,462 genotypes.

We detected ROHs using DARWINDOW (https://github.com/mennodejong1986/Darwindow; [50]). We used a sliding window size of 25-kb, a minimal window number of 4 and the heterozygosity threshold was set to 5%. The inbreeding coefficient $F_{ROH}$ was calculated as the proportion of the genome covered by ROH [51]. Identified ROH were classified according to their length in different bins for each group. We used again a double-sided Bayesian paired t-test implemented in the R package *BayesianFirstAid* [48] to assessed differences between $F_{ROH}$ of the different groups.

## Function of selected genes and selection on protein coding genes

For $F_{ST}$ above 99.8% CI between Hawaiian feral *vs* domestic cats and Australian feral *vs* domestic cats independently, we obtained gene IDs for putatively selected genes from *Felis catus* annotation genome file. We then retrieved putative gene functions associated with these IDs from the UniProtKB (https://www.uniprot.org/; accessed 2.3.2022) and also searched for KEGG (Kyoto Encyclopedia of Genes and Genomes) pathways associated to these genes in the KEGG pathway database (https://www.genome.jp/kegg/pathway.html; accessed 2.3.2022). With a not too large number of differentially expressed genes (DEGs) candidates, manual scanning of the comprehensive knowledge united in the UniProt database can yield literature-based insights on their functional or other relevant communalities [52]. For this, we manually scored the respective UniprotKB entries for DEGs in a two-step process. In a first step, genes are screened to define their communalities, and then we did a second screening to consistently assign each gene to a then well-defined category. Based on the abundance of these genes we defined the following categories: nervous system development, musculoskeletal system,

reproduction, DNA repair, response to heat/UV radiation, nutrition, immune response or not fitting to any of these categories.

### Analysis on divergent windows

In order to identify which selective processes might have potentially acted on the divergent windows previously identified, we filtered the intersection between the $F_{ST}$ (above 99.8% CI) between each of the feral cat group and domestic cats with π and $T_D$ values that were in the lower 5% quantile of the respective population for each of the feral cat groups. Following Feulner et al. [53]and Pfenninger et al. [54], we considered four different selective processes in 100-kb divergent windows by examining π and $T_D$ of feral cat groups in comparison to domestic cats. We classified the scenarios as follows: (i) low $T_D$ and π in derived feral population and inconspicuous $T_D$ and π domestic cat population, indicating selective sweep or positive selection in feral populations; (ii) inconspicuous $T_D$ and π in feral populations and low inconspicuous $T_D$ and π in domestic population, indicating relaxed selection in the feral populations; (iii) low $T_D$ and π in both, feral and domestic populations, indicating background (or purifying selection) and; (iv) inconspicuous $T_D$ and π in both, feral and domestic populations, indicating divergence driven random drift due to reduction of gene flow (S2 Table in S1 File).

## Results

### Whole-genome sequencing and population structure

We generated a whole-genome sequencing dataset for the 27 individuals of a mean coverage of 23X. A total of 20,649,872 SNPs were identified and 15,566,275 bi-allelic SNPs were kept in the full dataset after passing filtering process. After pruning for LD 1,899,534 unlinked SNPs were kept.

The PCA approach clearly clustered individuals into different groups (Fig 1b). PC1 explained 18.3% of the total variance and split wildcat individuals and *Felis catus* individuals (feral + domestic cats). PC2 explained 7.9% of the total variance and split Hawaiian feral cats and the rest of *Felis catus* individuals. Within this latter group, Australian feral cats clustered with North American domestic cats, while other subgroups were observed (Far East, Middle East and European domestic cats). The highest supported number of genetic clusters (*K*) was *K* = 2 (CV error = 0.64497; S1 Fig in S1 File), followed by *K* = 4 (CV error = 0.72890) and *K* = 3 (CV error = 0.80142). For *K* = 2 all Australian feral individuals and only one Hawaiian feral cat individual showed admixture between the two genetic pools (Fig 2a). ADMIXTURE results identified structure amongst the feral cat groups and North American and European domestic cats.

The ML analysis showed high support for all major relationships (BS > = 98%, Fig 2b). Hawaiian and Australian feral cats formed monophyletic clades, both with BS = 100%. Hawaiian feral cats appeared sister to a clade of North American domestic cats, while Australian feral cats appeared sister to a clade formed by North American and European domestic cats.

The mean genome-wide $F_{ST}$ between feral and domestic cats was 0.0464416. We identified a total of 487 SNPs with $F_{ST}$ above the 99.8% threshold for Australian and Hawaii feral cats with domestic cats ($F_{ST}$ = 0.358924 and $F_{ST}$ = 0.4300788 respectively; Fig 3a). Genes containing SNPs above threshold were considered for further analyses. Absolute divergence ($D_{XY}$) was 0.219 between Australian feral *vs* domestic cats, and 0.218 between Hawaiian *vs* domestic cats (Fig 3b).

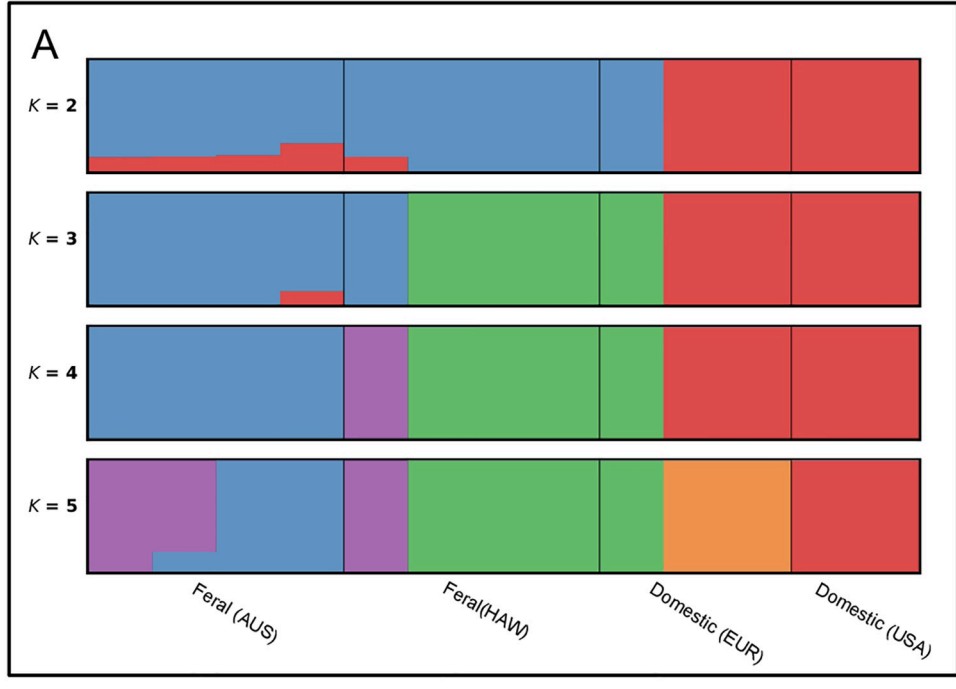

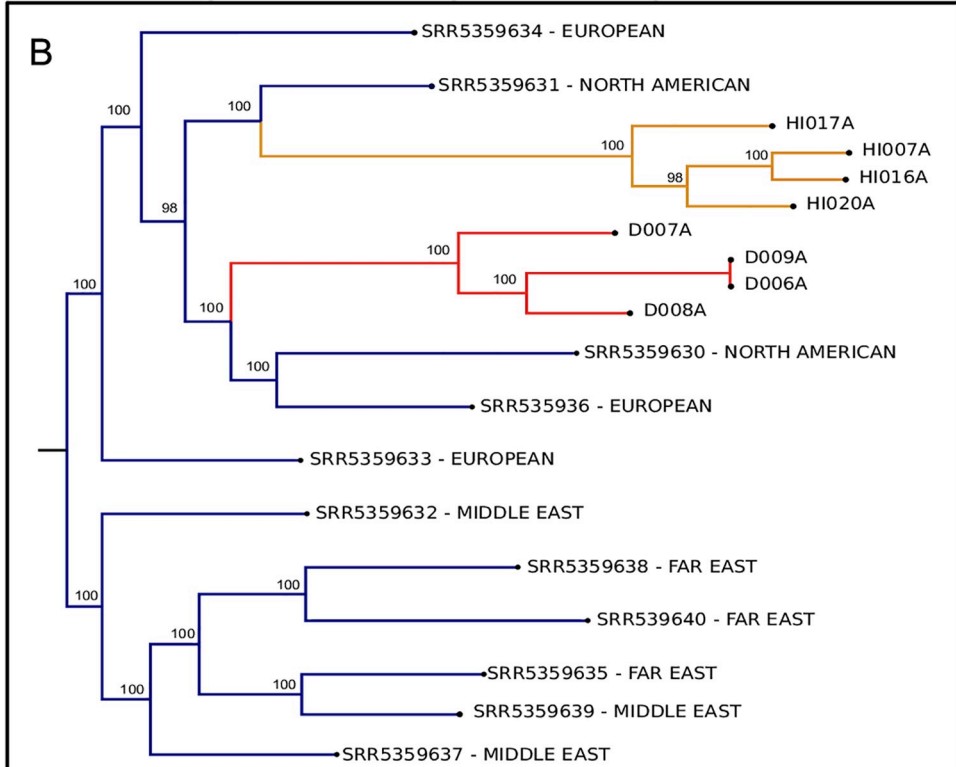

**Fig 2.** a) ADMIXTURE plot showing ancestry for each individual belonging to *K* 2–5 genetic clusters, b) Maximum Likelihood tree with colours indicating the origin of each individual (colour descriptions as in b). Numbers above nodes show bootstrap values.

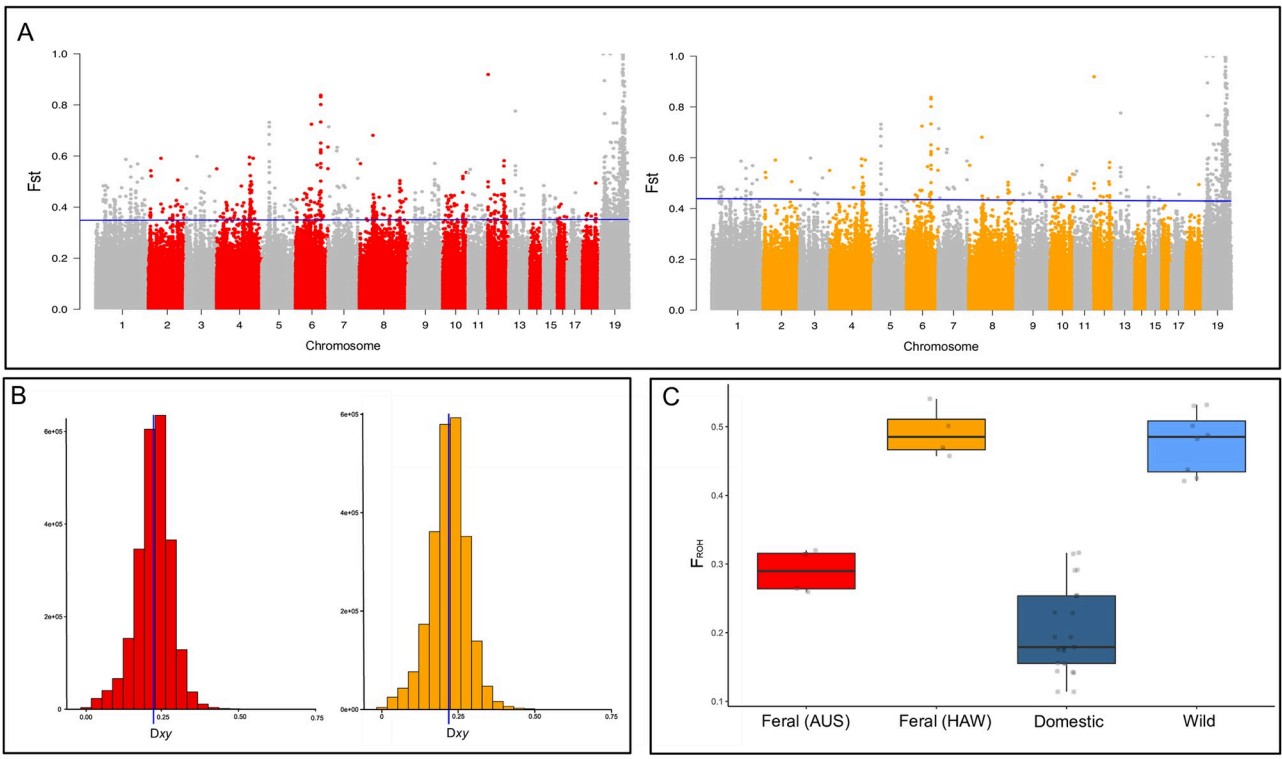

**Fig 3.** a) Manhattan plots of $F_{ST}$ in 10-kb non-overlapping windows with the 98% $F_{ST}$ threshold for Australian feral cats (in red) and Hawaiian feral cats (in orange); b) Distribution of absolute divergence values between Australian feral and domestic cats (in red) and Hawaiian feral and domestic cats (in in orange); c) Box plots of the inbreeding coefficients inferred from runs of homozygosity indicating the distribution of the per-individual number of ROH in different populations. Points indicate individuals.

### Inbreeding in feral cats

Wildcats and Hawaiian feral cats showed similar inbreeding coefficients ($F_{ROH}$ = 49%), while Australian feral cats showed intermediate levels of inbreeding ($F_{ROH}$ = 30%). Domestic cats showed the lowest inbreeding levels ($F_{ROH}$ = 21%) out of all groups (Fig 3c). Furthermore, based on the ROH lengths distribution (S2 Fig in S1 File), Hawaiian feral cats showed recent inbreeding as seen by the frequency of long ROHs (53.3% of ROHs > 5 Mb). On the contrary, wildcats showed that the acquisition of the ROH occurred rather long ago (57.3% of the ROHs < 1 Mb). The total coverage of the genome by ROH of Australian feral cats and domestic cats was lower than both of the aforementioned groups and evenly distributed along the different bin lengths. Pairwise Bayesian t-test showed that all observed differences amongst cat groups were different from zero with absolute certainty (S3 Fig in S1 File).

### Genetic diversity and demographic history inference

π was generally very low for each cat group (Fig 4a). Domestic cats show higher π than feral congeners and, wildcats showed the lowest π.

Mean $T_D$ showed slight differences in mean values amongst cat groups, with feral cats exhibiting higher $T_D$ and wildcats the lowest (Fig 4b). Mean $θ_W$ estimates were also higher in domestic cats than their feral congeners and, wildcats showed the lowest $θ_W$ (Fig 4c and S4 Fig in S1 File).

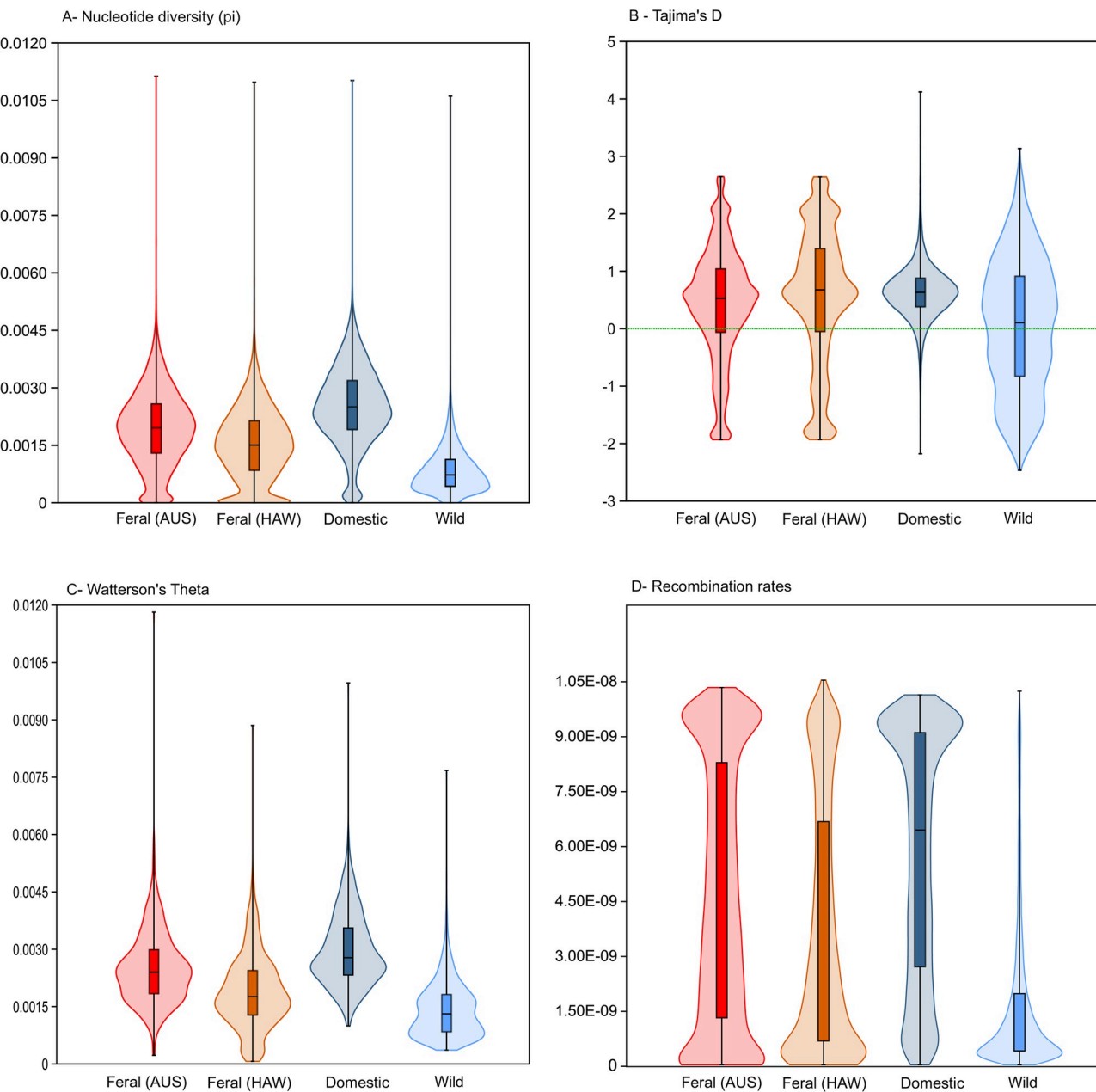

**Fig 4.** a) Nucleotide diversity (π) for each cat group; b) distribution of Tajima's *D* (T_D) for each cat group; c) Watterson's theta (θ_W) distribution for each cat group. All distributions used 100-kb windows; and d) Average recombination rates for feral and European domestic cats (c/bp = centimorgan per base pair). Colour coding as described in Fig 1b.

## Recombination rates amongst cat groups

Average recombination rates for each group (feral and European domestic cats) are shown as violin plots in Fig 4d. European domestic cats show the highest recombination rates (6.03E-09 c/bp; 95% HDI 4.71E-09-7.09E-09) followed by Australian feral (4.91E-09 c/bp; 95% HDI 3.7E-09-6.02E-09) and Hawaiian feral cats (3.84E-09 c/bp; 95% HDI 2.68E-09-5.03E-09). Wildcats show the lowest recombination rates of the four groups (1.65E-09 c/bp; 95% HDI

9.57E-10-2.54E-10). Training and validation data converged approximately after 25–60 over time (epochs). Mean absolute error and mean squared error of raw predictions, convergence of loss (measured by mean squared error) over time during training and distribution of parametric bootstrapping predictions can be found in S5 Fig in S1 File. Confidence intervals for the predictions are shown in S6 Fig in S1 File for each chromosome per group. For all three groups, chromosome number 19 (the sex chromosome) showed the lowest recombination rate. Pairwise Bayesian t-test showed that all observed differences amongst cat groups were different from zero with absolute certainty (S7 Fig in S1 File).

## Highly differentiated genes and processes driving differentiation in feral cat

We found 105 and 94 highly differentiated genes for Australian and Hawaiian feral cats and domestic cats, respectively (S3 Table in S1 File). For Australian feral cats, almost 30% of these genes were related to nervous system development, followed by 13% and 12% reproduction and musculoskeletal system, respectively (Fig 5a). For Hawaiian feral cat almost 30% were also related to nervous system development, followed by 13% and 12% related to reproduction and immune response. Twenty genes were found in both populations, six of them related to nervous system development; two related to musculoskeletal system, three related to immune response, one related to reproduction and one related to DNA repair (Fig 5a). For seven highly differentiated genes found in both datasets we could not find proper annotation information.

We found 218 and 152 highly divergent 10kb windows, for Australian and Hawaiian feral cats respectively, for which $\pi$ and $T_D$ fell within the lower 5% quantile (Fig 5b). Most highly divergent regions Australian feral cats evolved by drift (177 regions; ca. 81%; Fig 5c), followed by selective sweeps and relaxed selection (20 [ca. 9%] and 19 regions [ca. 8%], respectively; Fig 5c). In Hawaiian feral cats also most highly divergent regions evolved by drift (130 regions; ca. 85%; Fig 5c) and relaxed selection played a substantial role (18 regions; ca. 12%; Fig 5c). Only a few regions in Hawaiian feral cats showed signs of selective sweeps (4 regions; ca. 2%; Fig 5c). Background selection seems to be negligible for both feral cat populations (2 regions and none for Australian and Hawaiian feral cats respectively; Fig 5c).

## Discussion

The four individuals from the two feral cat populations serve as a valuable proof-of-principle study. Despite the distinct differences between these feral populations, we identified common patterns in their evolutionary trajectories post-feralization, particularly concerning the categories of highly differentiated genes and the evolutionary processes that influence feralization. $F_{ST}$ analyses can be estimated precisely with a small sample size, as small as $n = 3$–6, when a large number of bi-allelic genetic markers are used [55, 56]. Nucleotide diversity statistic ($\pi$) is descriptive only, while the Watterson's theta ($\theta_W$) gains most of its information from the first individuals added. Tajima's $D$ ($T_D$) is derived from the frequency-spectrum inference, and only inferences of very recent events require large sample sizes [57]. In the same line, the deep learning method used in this study to infer genome-wide recombination rates can be used with as few as four individually sequences chromosomes [49]. The ROH estimation method we used works at individual level, although estimations are later averaged. Even though it is important to interpret our findings with caution given the limited sample size of this study we are confident that our dataset allows us to draw preliminary conclusions and generate new hypotheses.

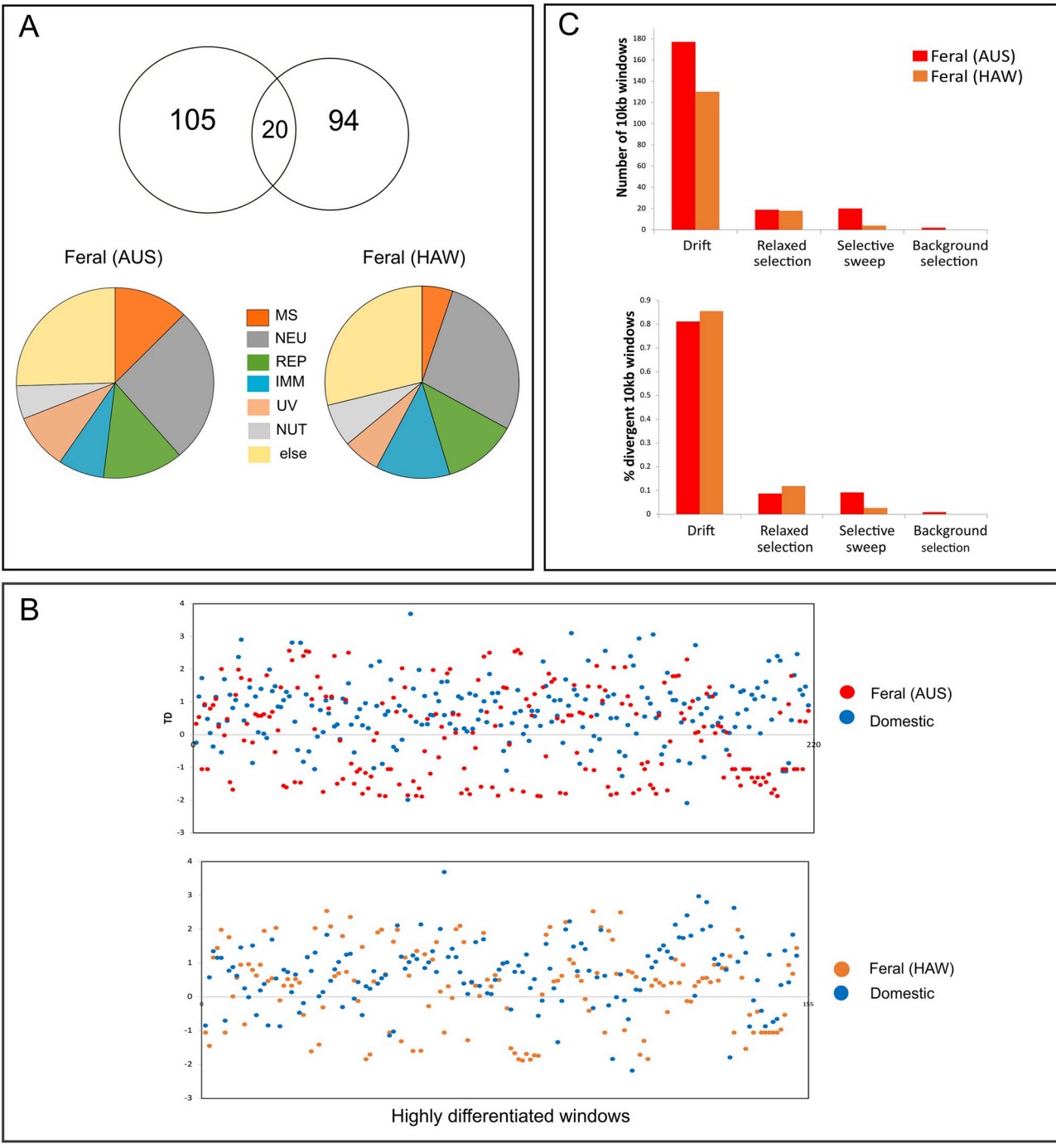

**Fig 5.** a) Highly differentiated genes between feral and domestic cats obtained from $F_{ST}$ above 99.8% CI (MS = musculoskeletal, NEU = neurological, REP = reproduction, UV = UV radiation, NUT = nutrition, IMM = immune system and others (ELSE); b) signature of four evolutionary processes in 100-kb windows containing at least one highly divergent 10-kb FST outlier window of feral cat populations when compared to the domestic cat population; and c) $T_D$ at highly differentiated regions between Australian feral *vs* domestic and Hawaiian feral *vs* domestic cats, respectively.

## Clear population structure and low genetic diversity of feral cats

Analyses of population structure identified *Felis catus* individuals (domestic + feral cats) and *Felis silvestris* (wildcats) as genetically distinct populations, consistent with previous studies [58, 59]. Within domestic cats, we observed different, well defined groups (Far East and Middle East, European and North American; Fig 1b). As suggested by previous studies a European ancestry for both islands cat populations can be assumed [60, 61] as a result of accidently or purposely introductions of ship-cats from vessels travelling the global maritime trade routes in the beginning of the 19th century [60, 62]. The finding of a common source is supported by a previous phylogeographic study [60] where shared haplotypes between the Dirk Hartog Island and Kaho'olawe cat population were found. In the light of the present results, it appears likely that the two feral cat populations stem from similar ship introductions with global routes spanning from North America to Europe and Australia [60, 61].

Between the two feral populations however, the Hawaiian population was substantially more diverged from the putative North American source population. This hinted either at a different timeframe for their introduction, stronger subsequent isolation and/or a stronger demographic bottleneck. The first connections between North America and Hawaii were established in the 1820s [63], thus giving an approximate time-frame for the initial introduction of cats from there. The close relationship of the Australian island population to the North American continent suggested a relatively recent origin. Since Dirk Hartog Island was discovered by a Dutch sea captain in 1616 (Dutch East India Company) and subsequently reportedly visited by French, English and Dutch explorers [62], the North American connection of the feral cat population remains so far historically unexplained.

We observed a reduction of $\pi$ in feral cats in comparison to the domesticated individuals (Fig 4a) which suggested that during the process of feralization, as it happened often during domestication [64], there is a decrease of genetic diversity. The observed reduction in $\pi$ likely reflects founder effects, stemming from the limited number of cats initially introduced, rather than solely being a consequence of the feralization process itself. Differences in the level of $\pi$ reduction in Australian and Hawaiian feral cats were expected, taking into account the potential different timelines and unknown overall number of domestic cat introductions, the degree of genetic drift and levels of potential introgression in both populations. Indeed, Australian feral cats showed higher levels of genetic diversity which could be due to ongoing gene flow with cats from rare and occasional releases resulting from the ecotourism activities in Dirk Hartog Island.

Interestingly, both feral cat populations show nevertheless a higher level of $\pi$ than the sample of European wildcats. However, including samples from a wider range of European wildcat populations could potentially alter the observed patterns. The low $\pi$ in wildcats is in line with previous findings where it has been shown that European Central wildcats went through severe bottlenecks in the late 19th and early 20th century [59, 65]. Our results of a mean $T_D > 0$ (Fig 4b) indicated an excess of intermediate alleles and lack of rare alleles. Empirical and simulated studies have shown genome-wide positive $T_D$ as result of bottlenecks [64]. Nevertheless, the positive $T_D$ observed in our data it is most likely the result from founder effects where the initial small population size and genetic drift increase the proportion of these alleles. A positive $T_D$, could also reflects long-term events, where genetic traits that were selected during domestication might have been neutral or slightly deleterious in the wild ancestors. Because of this, these traits were present in the wild population due to a balance between random genetic changes and natural selection. This historical balance, combined with the selective pressure of domestication, can lead to an excess of intermediate alleles [66].

The observed differences of inbreeding in Australian and Hawaiian feral cats were also expected. The lower levels of inbreeding observed in Australian feral cats compared to Hawaiian suggests the possibility of more frequent gene flow, perhaps due to rare and occasional releases mentioned above. In the same line, admixed groups tend to have fewer ROHs [67], which is consistent with the larger number of admixed individuals observed for feral populations (Fig 2a). We observed more recent inbreeding in Hawaiian feral cats, as indicated by the larger frequency of long ROHs (S2 Fig in S1 File) in comparison to Australian suggesting that other factors might be playing a role in the levels of inbreeding of feral populations. Some of these factors include the different sources of domestic cats of feral populations, a stronger effect of genetic drift, bottleneck event or strong founder effect due to the small number of individuals establishing the population. Our results point to low levels of inbreeding in domestic cats, which challenges the general assumption that domesticated species show considerable inbreeding [68, 69]. However, it is important to consider that domestic cats may also result from the admixture of various breeds, which can lead to higher genetic variability and consequently lower levels of inbreeding. Also noticeable is the high inbreeding coefficient of wildcat in comparison to previous studies on European wildcats [35, 70]. In addition, we observed a considerable frequency of short ROHs in wildcats indicating ancient inbreeding, which is consistent with the fact that over time long segments in the chromosomes will have a tendency to breakdown due to recombination [71]. An alternative explanation to the frequency of short ROHs in wildcats could be severe bottlenecks in the past and previous centuries in European wildcats [59, 65]. Our findings should be interpreted with caution due to the limited sample size, which can potentially affect the accuracy of ROH estimates and reduce the power to detect selection signatures.

## Higher recombination rates in domestic cats

Recombination rates are influenced by genetic, epigenetic, and environmental factors, and are highly variable at multiple scales, for example between species, populations of the same species, individuals of the same population as well as across different regions of a chromosome and sexes [72, 73]. It is expected that recombination rates increase in domesticated species, driven by strong artificial selection [74]. However, the increase of recombination rates in domesticated species in comparison with their wild counterparts has been observed in birds [11], insects [75] and numerous plant species [76], but up to now not for mammals [77]. There has been little prior evidence or predictions of how the feralization process influences recombination rates. Our results show convincing genome-wide evidence of strongly reduced recombination rates in both independently derived feral cat populations. This could suggest that the absence of strong artificial selection pressure, such as domestication, might generally tend recombination rates to decrease, even in the presence of novel environmental selection pressures upon returning to the natural conditions. Direct cytological observation of cross-overs in individual sperms in Muñoz-Fuentes et al. 2014 [78] yielded different results than the inference from genome-wide genotypes here. This might be explained by the fact that the latter integrates the results of different demographic histories [79], selection [80] and drift [81] over many generations.

Within the feral cats, the Australian island cats showed higher recombination rates than the Hawaiian. If feralization relieves the selection pressure on recombination rate, the Australian population might not yet have had enough time for a more substantial decrease. Additional explanations could include varying selection pressures between islands or the continuous gene flow of Australian feral cats with domestic cats, as discussed in the previous section. Although there is no clear consensus on the relationship between environmental factors and

recombination rates, there are examples showing that this relationship resembles either a U-shaped curve [78] or a reversed U-shaped curve on mammals. In particular, extreme and fluctuation in temperature has been shown to impact recombination rates [82–84]. We hypothesized that higher recombination rates observed in the Australian feral cats could be due to the higher mean daily temperatures to which they are exposed along with higher solar radiation. However, our study did not explicitly test these aspects and this requires further investigation.

## High differentiation in different genomic regions in feral populations

The genomes of Hawaiian and Australian feral cats show substantial differentiation to their domesticated ancestors and molecular traces of evolutionary processes causing a feral state. Although the feralization process seemed to follow similar routes in the two instances, the prevailing forces and targets were slightly different for each population. The overall great functional similarities of the genes affected clearly pointed to a similar change in selective regimes, yet different genes were involved. Both populations showed highly differentiated genes related to nervous system development, reproduction and various other categories (immune response, nutrition, DNA damage and response to heat and UV radiation).

The strong differentiation in genes related to DNA damage and response to heat and UV radiation may be explained by the generally high radiation levels on both islands with Australia having exposure levels in the extreme (11+) range (https://wmo.int/topics/climate; accessed 24.6.2022) [85].

Around 30% of differentiated genes related to nervous system development and up to 13% related to reproduction in both populations are a strong indication how relaxed selection impacts the nervous system and reproductive genes. Reproductive protein genes have been shown to diversify faster than most other gene categories, especially if involved in reproductive processes [86]. The appearance of mutations related to novel traits have been linked to a relaxation of natural selective pressure in domestic populations [87]. Vice versa we expect relaxed selection pressure to promote traits of the nervous system that allow refinement of skills important for the survival in wild (e.g. hunting skills).

The most striking difference between the two populations was the function of genes related to musculoskeletal system. While the genomes of Australian feral cats showed 13 highly differentiated genes with this function, only five were found for the Hawaiian feral cats. This might have been due to different prey spectra. Prey species numbers differ extensively between the islands with the Australian Island having a large variety and amount of possible prey species (invasive house mice, lizards, birds etc., [32]) compared to the Hawaiian Island [88–90]. Here, the main food sources were insects, invasive house mice and a few bird species. Differences in the health of the cat populations are also apparent in the overall mean weight measurements recorded during the respective trapping periods, where Australian feral cats displayed higher mean weights than Hawaiian feral cats (Dr. K. Koch personal communication). On the basis of the distinctions in their overall weight, the high number of differentiated genes related to the muscoskeletal system in Australian feral cats might give a genetic basis, which could reflect the differential prey spectrum. These differences in an overall remarkably similar evolutionary response may highlight the potential to response to the particular environmental characteristics of the two islands. Despite the extended notion that feralization could be considered as the reverse of domestication our study does not support this view. Even though there are evidences of reversal of traits, these changes might be due to novel genetic changes [2, 91, 92]. A population genomic study on feral pigs suggested that feral forms are closer to wild pigs, based on genomic signatures of natural and purifying selection [7]. Observed morphological similarities between wild pigs and feral animals were not due to the reversal through gene flow, but instead

the result of unique adaptations by feralization [7]. Even though the genetic background of feral, domestic and wildcats lies further in the past compared to pigs, our study shows analogous flexibility in the genome in the adaptation to a feral state.

## Processes driving feral and domestic cat's differentiation

Our findings show that similar evolutionary regimes played a role in the differentiation of feral and domestic cats. As expected, the effect of random drift was extensive (Fig 5b), potentially due to population isolation and lack of gene flow [93] in addition to the small population sizes following the founder events of both feral populations [94–96]. This pattern has been observed in Island fox (*Urocyon littoralis*) of the Channel Islands, where genetic drift appears as the dominant evolutionary mechanism driving population divergence among island fox population [97]. In the case of Dirk Hartog Island and Kaho'olawe gene flow with further sources of domestic cats can be considered absent to very low after the initial introductions of cats, since both islands were only temporary inhabited for farming settlements, leaving cats to evolve to the feral state [60]. Much larger islands or continental populations would have potentially reduced the effect of random drift in favoured of positive selection as shown for other mammal species such as Canadian lynx (*Lynx canadensis*; [98]). The explanation as to why random drift played a slightly bigger role in Hawaiian feral cats (85% of highly divergent regions) in comparison to in Australian feral cats (81% of highly divergent regions; Fig 5c) remains unclear, however we hypothesized that this might be due to the stronger effect of random drift in smaller islands, differences in their demographic histories and/or a stronger bottleneck effect in Hawaiian feral.

More interestingly, we provide evidences of relaxed selection on both feral populations, as indicated by the negative $T_D$ in domestic but not feral cats (Fig 5b), presumably as result of the removal or weakening of selective pressures domestic populations must have faced, as it has been already hypothesized for other species [99] and feral species in particular [6, 92, 100]. This lack of rare alleles in the feral populations could again be explained by the small population sizes of feral populations. In contrast to the above patterns, we would expect excess of rare alleles and negative $T_D$ [101] in the feral populations, but not in the domestic ones, if feral cats experienced selective sweeps upon returning to the wild. Selective sweeps have been detected in Hawaiian feral chicken [5, 6] and feral sheep [102] and positive selection in feral dingoes [12]. We found regions evolving under selective sweeps in both feral cat populations (9% and 2% of all highly divergent regions for Australia and Hawaiian, respectively). The minor role selective sweeps played in comparison to drift and relaxed selection in Hawaiian feral cats indicates the absence of strong positive selection in feral cats upon facing novel selection pressures upon escaping domestication. Similarly, we would expect to see negative $T_D$ in both groups if the highly divergent regions evolved under background selection [103]. However, we did not detect this pattern in Hawaiian feral cats and just two divergent regions (0.9% of all highly divergent regions) in Australian feral cats.

## Conclusions

This study demonstrated that feralization of cats is an excellent model to explore the evolutionary processes acting after the release of mammal populations from human control. The overall trend of our study shows that feralization in cats is a complex evolutionary process that brings feral cats on a unique evolutionary trajectory with an open outcome. This study shows clear indications of major changes in the genome through feralization. The most surprising finding was that adaptive evolution played a minor role compared to the relaxation of the domesticated state. Further sampling and studies, including the examination of stray cats, are needed

to differentiate genes or gene combinations favored by feralization, depending on the specific environmental and demographic contexts of their release from human control. Additionally, the inclusion of domestic cats ancestor *F.lybica*, would assist investigating highly differentiated genes across wild, domestic, and feral cat populations and provide a broader understanding of the domestication and feralization continuum. This comprehensive approach will enhance our knowledge of the genetic mechanisms underlying these processes and how they are influenced by varying conditions.

## Supporting information

**S1 File.**
(DOCX)

## Acknowledgments

The authors would like to dedicate this study to the memory of Dr. Fern P. Duvall II.

## Author Contributions

**Conceptualization:** María Esther Nieto-Blázquez, Markus Pfenninger, Katrin Koch.

**Formal analysis:** María Esther Nieto-Blázquez, Manuela Gómez-Suárez, Markus Pfenninger.

**Investigation:** María Esther Nieto-Blázquez, Manuela Gómez-Suárez, Markus Pfenninger, Katrin Koch.

**Supervision:** Markus Pfenninger.

**Validation:** María Esther Nieto-Blázquez, Markus Pfenninger, Katrin Koch.

**Visualization:** María Esther Nieto-Blázquez, Katrin Koch.

**Writing – original draft:** María Esther Nieto-Blázquez.

**Writing – review & editing:** María Esther Nieto-Blázquez, Manuela Gómez-Suárez, Markus Pfenninger, Katrin Koch.

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
