## [Decision Letter · Decision Letter 0]

16 May 2024

PONE-D-24-04531Impact of feralization on evolutionary trajectories in the genomes of feral cat island populationsPLOS ONE

Dear Dr. Nieto Blázquez,

Thank you for submitting your manuscript to PLOS ONE. After careful consideration, we feel that it has merit but does not fully meet PLOS ONE’s publication criteria as it currently stands. Therefore, we invite you to submit a revised version of the manuscript that addresses the points raised during the review process.

We look forward to receiving your revised manuscript.

Kind regards,

Franck Courchamp

Academic Editor

PLOS ONE

Journal Requirements:

The research was partially funded by the Landes-Offensive zur Entwicklung Wissenschaftlich ökonomischer Exzellenz (LOEWE) Program of the Hessian Ministry of Higher Education, Research, Science and the Arts through the LOEWE Centre for Translational Biodiversity Genomics (LOEWE-TBG)

3. Thank you for uploading your study's underlying data set. Unfortunately, the repository you have noted in your Data Availability statement does not qualify as an acceptable data repository according to PLOS's standards.

Additional Editor Comments:

The authors aimed to identify the effect of feralization in two distinct feral cat populations on insular ecosystems (in Australia and Hawaii), analysing the genome of 4 samples each and compared them with the available genome of 11 domestic cats (n=11) widespread throughout the world, and with 8 wildcats from Germany. The results suggest that feralization in cats is not just the reversal of domestication, but an independent and more complex process. This is an interesting and well-written manuscript.

I have collected the professional opinion of four different experts on the topic and they recommend Major Revision (3) and Minor Revision (1); I therefore recommend Major Revision. I urge the Authors to seriously take into account the comments, as Major Revision does not guarantee acceptance of the revised version. I hope the comments of the reviewers will be useful.

Reviewers' comments:

Reviewer's Responses to Questions

**Comments to the Author**

1. Is the manuscript technically sound, and do the data support the conclusions?

Reviewer #1: Yes

Reviewer #2: Partly

Reviewer #3: Partly

Reviewer #4: Partly

2. Has the statistical analysis been performed appropriately and rigorously? 

Reviewer #1: Yes

Reviewer #2: Yes

Reviewer #3: Yes

Reviewer #4: N/A

3. Have the authors made all data underlying the findings in their manuscript fully available?

Reviewer #1: Yes

Reviewer #2: Yes

Reviewer #3: Yes

Reviewer #4: Yes

4. Is the manuscript presented in an intelligible fashion and written in standard English?

Reviewer #1: Yes

Reviewer #2: Yes

Reviewer #3: No

Reviewer #4: Yes

5. Review Comments to the Author

**Reviewer #1:** In this paper, Nieto-Blázquez ME et al. presents an interesting and well-written study on the feralization processes that might have independently occurred in two geographically distant feral cat island populations. By analyzing whole-genome sequences data of reference domestic and European wildcats and feral cats from Dirk Hartog Island and Kaho’olawe Islands, the authors exhaustively demonstrated that the feralization of cats is a complex evolutionary process and not simply a reversal of domestication, and that random drift and relaxed selection were the prevailing forces that likely have driven the major changes in the genomes of feral cats compared to domestic cats.

The manuscript will be eligible for publication in the PLOS ONE Journal after making a few minor suggestions.

• Highly differentiated genes and processes driving differentiation in feral cats: It would be interesting to even identify highly differentiated genes for Australian/Hawaiian feral cats and European wildcats to verify the presence of genes specifically adapted to the wildlife (such as genes related to socialization, metabolism, fear, digestion). For example, Zhang et al 2020 identified several genes in regions under selection in dingoes that were more similar to gray wolves than to domestic dogs, suggesting that selection on such genes occurred in the dog lineage after the split from the dingo ancestors. If this is not possible, it would be useful to insert a few cautions in the Discussion paragraph or suggest these analyses as prospects.

• Demographic analyses: I have some doubts about the use of the PSMC model which allowed the authors to investigate past and not recent events. The feral populations analyzed derive from domestic cats imported about 200 years ago on the islands. Therefore, the PSMC results showing Ne decrease patterns with a peak at 500k years ago are not interpretable, considering that domestication itself dates back no earlier than 10-15k years ago. The authors should explain the choice of this analysis, and the results obtained, and list the cautions in the Discussion part.

• Figures: The authors should improve the resolution of the figures in the main text (in particular in Fig.1c the writings are illegible), and standardize the names and codes of the four populations in all figures and tables.

• Line 432-434: This hypothesis is not clear to me. I would suggest the authors explain it better.

• Line 377-389: I would suggest the authors start the paragraph with "reliable conclusion and new ....even though we acknowledge ...", and then list the various cautions. Furthermore, I would suggest writing the paragraph more organically so that the various sentences do not seem like a list of answers to specific questions.

Decision: accepted with minor revisions

**Reviewer #2:** The authors aimed to identify the effect of feralization in two cat populations inhabiting Dirk Hartog Island [Australia] and Kaho’olawe [Hawaii], respectively. The authors analysed the genome of 8 samples (4+4) and compared them with the available genome of domestic cats (n=11) widespread in Europe, America and Asia, and with wildcats (n=8) collected only from Germany, where the population suffered a strong decline.

Genomic analyses were correctly performed but I found critical issues in sampling, in some Results, and consequently, in the Discussion. I suggest reconsidering the analyses of more reference samples and avoiding analyses negatively influenced by the low sample size.

Moreover, the authors should consider softening some assumptions: e.g. the absolute absence of releasing of domestic cats after 1800. Particularly, Dirk Hartog Island is interested in ecotourism, so it can be hypothesized in the discussion the possibility that some results have been affected by rare and occasional cat releases. Higher caution should be used in interpreting the data from GWAS, as several variables could influence the results, and reference samples (both domestic and wildcats) are from very different habitats. It is hard to define which factors have been influencing adaptation events when different variables are acting differently in the studied groups.Some further comments:

P5 L123 The authors wrote: “We included four feral cat samples per island”. As declared by the authors at the beginning of the Discussion, a sampling of only 8 feral cats can negatively influence the results.

P5 L124 The authors wrote: “Both islands are not inhabited by humans and, domestic cats were introduced in the last 200 years”. This affirmation is not completely true as the Dirk Hartog Island [Australia] is known as an ecotourism destination.

P5 L129 The authors wrote: “indicating similar environmental regimes”. There are several similarities in the ecological conditions of the two islands that authors recorded; however, Dirk Hartog Island [Australia] is quite three times larger than Kaho’olawe [Hawaii] and is human inhabited.

P5 L130 The authors wrote: “included 8 German wildcat individuals”. Comparison with samples originating only in Germany can affect the bias. It must be considered that some features could be different in animals inhabiting island habitats and semi-arid climates. Differences in genetic variants could be recorded when analysing individuals inhabiting very different habitats. I suggest the introduction in the analysis of wildcats’ genomes from similar ecological areas; if it is not possible, I suggest adding adequate argumentations in the discussion of the results, particularly those referring to GWAS.

P6 L150 The authors wrote: “… and filtered for biallelic variants passing all filters”. I suggest adding information on applied filters, e.g. quality SNP and quality sample thresholds.

P6 L159 I suggest computing more K.

P11 L259 The authors should describe details about the genome data, e.g. reads, contigs, and so on.

P11 L260 The authors should describe the total amount of SNP identified and how many were removed after using the different filter steps (e.g. for SNP or sample quality).

P 11 L262 The authors wrote: “two main groups”. It seems that the main groups are three.

P11 L267-270 It is impossible to read the population names in the Admixture plot, so it is not possible to verify the genetic structure described in the text. The draft would benefit from the graphical view of the cv plot from 1 to 4K. CV values for K=3 should be reported. PCA describes 5 clusters, so I suggest computing a higher number of K and plotting the CV values to identify the best K.

P11 L271-274 The authors wrote: “Hawaiian feral cats appeared sister to a clade of North American domestic cats”. In the supplementary material, SRRS35359631 is from Maui, so from Hawaii. There is a mistake in the tree because the sample from Iowa has been reported twice. Considering the position in the tree and the lack of the sample from Iraq, probably there was a switch of sample names.

P12 L275 Fst was computed between feral and domestic cats. Although wildcat sampling is only from Germany, it is important to carry out Fst computation also between feral and wildcats. As told before, the analysis would benefit from the inclusion of further wildcat samples.

P13 L17 The draft would benefit from a description of the differences between groups.

P13-14 L326-327 The authors wrote: “… both feral cat populations had 2.93 (95% HDI 2.91-2.94), respectively 2.31 (95% HDI 2.29-2.32) times higher recombination rates than European wildcats”. This observation does not fit with the plot from Hawaii.

P14 L 332-333 The authors wrote: “Wildcats and Hawaiian feral cats showed similar inbreeding coefficients (FROH = 49%), while Australian feral cats showed intermediate levels of inbreeding (FROH = 30%)” There is a strong bias in sampling and low representativeness, so I suggest rephrasing.

P14 L335-338 Considering “Ceballos et al. 2018 Runs of homozygosity: windows into population history and trait architecture”, the situation is not so simple to describe and the ROH from Hawaiian samples could derive from inbreeding but also from the founder effect. RHO in wildcats could describe an occurred bottleneck, but as told before, this bottleneck could not be representative of other populations of wildcats. The situation for Australian feral cats is more similar to domestics, but it would be taken into consideration that human presence can have favoured a gene flow with additional domestic cats during these past two centuries.

P14 L345. Differentiate genes should also be described for wildcats. As told before, although there is a bias in the representativeness, the comparison should be done.

P16 Discussion. The discussion would benefit from reconsidering the comments to MM and Results. Particularly:

P 16 L392 The authors wrote: “Analyses of population structure identified unique genetic clustering between Felis catus individuals (domestic + feral cats) and Felis silvestris (wild cats), as expected from previous studies” This affirmation does not seem validated from the results.

P16 L395-397 The authors wrote: “Surprisingly, the North American domestic cat population appeared closest to both feral cat populations” The specimen gathering with Hawaii sample is from Maui”

P16-17 L396-404 Also by considering the results from this study, the European origin is quite clear also for the American cats. I do not think it is possible to affirm that the origin of feral cats is due to cats escaping from ships sailing only from America.

P16-17 L401-402 The authors wrote: “The finding of a common source is supported by a previous phylogeographic study [61] where shared haplotypes between the Dirk Hartog Island and Kaho’olawe cat population were found.” This finding is justified by considering the European origin of the American cats.

P17 L405-407 The cluster of Australian feral cats joins a clade including European and American domestic cats, so this assumption is not right referred to in the present study. Australian feral cats cluster with European and American domestic cats.

P17 L408-417 From this data it is impossible to highlight differences in the timeframe of the release of domestic cats on the two islands. The main similarity between American domestic cats and feral cats from Hawaii is unsupported, as the reference sample is from Maui.

P17-18 L418-422 First, it is important to highlight how this information could be biased from the low number size; moreover, I suggest highlighting mainly the founder effect rather than reduction due to feralization.

P 18 L423 The authors wrote: “taking into account the different timelines of domestic cat introductions” I did not find evidence of it in the results.

P18 L425-426 The authors wrote: “However, both feral cat populations show nevertheless a higher level of genetic diversity than the sample of European wild cats.” There is a bias in the sampling of wild cats. By choosing different reference populations, the results could be different.

P18 L431 The authors wrote: “At first sight, this suggested that feral cat populations are in a population contraction phase” I think it would be better to consider a founder effect rather than a bottleneck.

P18 L440 and L447 High frequencies of long ROHs identify small pops (Ceballos et al. 2018)

Figures There is a general bad resolution of figures.

Figure 1 C The names in Figure 1 C are not readable.

Figure 1 D There is a mistake in the names, as reported in the comments above.

Figure S2. PSMC plot I found µ in the legend and α in the figure

Figure S5 The values in the figures are not readable.

**Reviewer #3:** I would like to thank the editor for the opportunity to review this manuscript. It is an interesting study in which the authors use WGS methods to analyse the genetic diversity, genetic structure, and ancestry/origin of two feral cat populations on the islands of Hawaii and Australia. The manuscript is relevant and timely. Although I appreciate the content of the manuscript, I think there are some points that need to be improved. I hope the authors will find the following comments useful.

INTRODUCTION

The introductory section begins with an explanation of the feralization and domestication processes in many animal species (chickens, pigs, etc.), a paragraph on the spread of cats after their initial domestication from Felis lybica, and a sentence on the negative effects of feral cats on the local ecosystem. All the written information in the introduction section is important, but it would be better to include the information from the list of articles in “References” or add additional references that use genomic approaches in the study of feral cats, but in the absence of references for feral cats, use studies that include genomic analyses of wildcats and domestic cats. In addition, a sentence or two should be written about the phylogeny of the wildcat, as the aim of the study involves comparing the molecular patterns of feral cats with their ancestors. The introduction section could include brief information about SNP markers. Describe the problem of feral cats as an invasive species on islands, in this case Australia, in a little more detail.

MATERIALS AND METHODS

This is a small sample size for a research article, but it can be justified by the use of a larger number of genetic markers, which is explained in the Discussion section.

For phylogenetic reconstruction analyses, I would include not only the German wildcat population, but also individuals from other biogeographical groups of wildcats:

1. Mattucci, F., Oliveira, R., Lyons, L. A., Alves, P. C. & Randi, E. European wildcat populations are subdivided into five main biogeographic groups: Consequences of Pleistocene climate changes or recent anthropogenic fragmentation? Ecol. Evol. 6, 3–22. https://doi.org/10.1002/ece3.1815 (2016).

2. Mattucci, F. et al. Genomic approaches to identify hybrids and estimate admixture times in European wildcat populations. Sci. Rep. 9, 1–15. https://doi.org/10.1038/s41598-019-48002-w (2019).

Please check the spelling of software, in lines 148-149 the authors wrote v0.7.15/v2.2.1, then in line 157 - v.1.0.7. and in line 158 – version 1.3.0. 3. Please check again the Materials and methods section and pay attention to a consistent spelling of software names and versions.

RESULTS

Figure 1: In my opinion, there are too many results for one figure. I suggest separating Figures 1c and 1d in the new figure. Rotate Figure 1c for a better understanding of the ADMIXTURE results.

Lines 313 – 317: Please rephrase the sentence so that “Supplementary Fig S3” is not repeated three times.

Line 334: Please separate Figure 2c from Figures 2a and 2b. Add a new figure label and a new caption for Figure 2c and link them to the text. Or put the paragraph “Inbreeding in feral cats” before “Genetic diversity and demographic history inference”.

Lines 364-372: I suggest first explaining the results of Figure 4b and then of Figure 4c.

DISCUSSION

The authors have answered the research questions, but when reading the discussion, I noticed that despite the large number of results (figures, supplementary materials) a weak overview of the research results was given.

Line 379: Please use symbol for nucleotide diversity instead of “pi”. Once defined, please use the symbols consistently.

Lines 402-404: This sentence is a bit unclear: “In the light of the present results, it appears likely that the two feral cat populations stem from ships originating or on routes from North America.”, please rephrase.

Lines 408-409: The authors write that “Between the two feral populations however, the Hawaiian population was substantially more diverged from the putative North American source population.” In lines 395-396, the authors write: “Surprisingly, the North American domestic cat population appeared closest to both feral cat populations.” Please pay attention to the Results and the correct interpretation in the Discussion and the Abstract.

Line 412: Missing reference after “the 1820s”.

Line 426: Please check the text. In some parts of the discussion the authors write “European wildcats”, then it is written in the text as “European wild cats”. Please correct as “wildcats”.

Lines 491-492: Please check the instructions for authors for in-text citations, for websites. Also make sure that your reference list includes full and current bibliography details for every cited work, as this source is not listed in the References.

Lines 507-515: The authors should indicate in Materials and methods how the feral cat samples were collected in Hawaii and Australia (more details on sampling), as the differences in average weight measurements during the respective trapping periods are mentioned in the Discussion.

Lines 529-531: Something is missing in the sentence in which the authors mention the Island fox, the sentence is not complete.

Line 545: Delete one “be”.

REFERENCES

Please format the references according to the submission guidelines, as some of the references need to be corrected. Examples:

Line 608: Capital letters of authors' last names, while other references are lowercase.

Lines 632, 685, 777, 779: wrong format for DOI

I would like to ask the authors to read the manuscript again and correct the errors in the text and check the grammar.

**Reviewer #4:** The ms "Impact of feralization on evolutionary trajectories in the genomes of feral cat island populations" analyzes genetic structure and differentiation between domestic and feral cats in two feral cat island populations in Australia and Hawaii, as well as domestic cats and European wildcats, by comparing whole-genome sequencing data.

As expected and thoroughly discussed, feralization in cats is not just the reversal of domestication, but an independent and more complex process.

The work is well written and presented, although some language checking should be performed (see points below).

Despite the first paragraph of the Discussion acknowledges and justifies the choice of analyzing only 4 samples per island, this might be a weak point in the presented work, and caution should be taken in drawing results and conclusions.

Authors do not explain in M&M: How were the feral samples obtained? How samples were taken and what was taken from feral cats (DNA from hair, blood, tissue)?

As well, authors explain only later in the ms why the considered islands are not inhabited by humans but are inhabited by domestic cats become feral, they might explain this point earlier, and add a very brief island history/description.

Supplementary material might be simplified.

Specific suggestions:

Line 88: delete comma after "Although"

Check for superfluous commas in other parts of the ms.

M&M: Try not to repeat excessively We...We...We...

Line 236 onward: Check sentence

Line 293-295: Check sentence

Line 398 and Line 418: Not clear

Line 427: that instead of than

Line 475: "could be due to"

Line 468 onward: the Australian population instead of Hawaiian?

Line 491: delete "range"

Line 545 and Line 553: Avoid repetitions

6. PLOS authors have the option to publish the peer review history of their article (what does this mean?). If published, this will include your full peer review and any attached files.

Reviewer #1: No

Reviewer #2: No

Reviewer #3: No

Reviewer #4: No

---

## [Author Response · Author response to Decision Letter 0]

25 Jun 2024

Response to reviewers

Additional Editor Comments:

The authors aimed to identify the effect of feralization in two distinct feral cat populations on insular ecosystems (in Australia and Hawaii), analysing the genome of 4 samples each and compared them with the available genome of 11 domestic cats (n=11) widespread throughout the world, and with 8 wildcats from Germany. The results suggest that feralization in cats is not just the reversal of domestication, but an independent and more complex process. This is an interesting and well-written manuscript.

I have collected the professional opinion of four different experts on the topic and they recommend Major Revision (3) and Minor Revision (1); I therefore recommend Major Revision. I urge the Authors to seriously take into account the comments, as Major Revision does not guarantee acceptance of the revised version. I hope the comments of the reviewers will be useful. 

Response: We thank the editor for the opportunity to submit a revised version of our manuscript and reviewers for their comments/suggestions/corrections. We found their feedback to be fair, thoughtful and constructive. We have addressed each comment/concern appropriately, see below.

** Note that line numbers in our responses correspond to line numbers in the clean version of the manuscript**

Reviewers' comments:

Reviewer #1: In this paper, Nieto-Blázquez ME et al. presents an interesting and well-written study on the feralization processes that might have independently occurred in two geographically distant feral cat island populations. By analyzing whole-genome sequences data of reference domestic and European wildcats and feral cats from Dirk Hartog Island and Kaho’olawe Islands, the authors exhaustively demonstrated that the feralization of cats is a complex evolutionary process and not simply a reversal of domestication, and that random drift and relaxed selection were the prevailing forces that likely have driven the major changes in the genomes of feral cats compared to domestic cats.

The manuscript will be eligible for publication in the PLOS ONE Journal after making a few minor suggestions.

We thank R#1 for carefully reading our manuscript and making useful comments and suggestions.

• Highly differentiated genes and processes driving differentiation in feral cats: It would be interesting to even identify highly differentiated genes for Australian/Hawaiian feral cats and European wildcats to verify the presence of genes specifically adapted to the wildlife (such as genes related to socialization, metabolism, fear, digestion). For example, Zhang et al 2020 identified several genes in regions under selection in dingoes that were more similar to gray wolves than to domestic dogs, suggesting that selection on such genes occurred in the dog lineage after the split from the dingo ancestors. If this is not possible, it would be useful to insert a few cautions in the Discussion paragraph or suggest these analyses as prospects. 

Response: We thank the reviewer for this comment and agree that it would be interesting to explore these questions in future studies. However, we believe it would be more appropriate to do this comparison with F.lybica, the ancestor of F.catus, rather than with F. silvestris. In addition, we would like to point out that the main aim of the study is to look at different between feral vs domestic cats, rather than comparing feralization vs domestication. Therefore, we focused on the genome differences and processes that have impacted the two feral populations after domestic cat releases in these islands. We provided lists of highly differentiated genes between feral vs domestic cats and discussed gene categories in the Discussion.

Nevertheless, we have incorporated the reviewer’s suggestion as a future research direction in the Conclusions, indicating that a more in-depth study of highly differentiated genes in wild, domestic, and feral cats would provide a broader understanding of the processes of domestication and feralization continuum. 

• Demographic analyses: I have some doubts about the use of the PSMC model which allowed the authors to investigate past and not recent events. The feral populations analyzed derive from domestic cats imported about 200 years ago on the islands. Therefore, the PSMC results showing Ne decrease patterns with a peak at 500k years ago are not interpretable, considering that domestication itself dates back no earlier than 10-15k years ago. The authors should explain the choice of this analysis, and the results obtained, and list the cautions in the Discussion part. 

Response: We thank the reviewer for this comment. After some thoughts, we agree that this analysis is not appropriate to answer any of the study questions. Given, that our PSMC results are similar to those in Nieto-Blazquez et (2022), we have decided to remove this analysis from the manuscript.

• Figures: The authors should improve the resolution of the figures in the main text (in particular in Fig.1c the writings are illegible), and standardize the names and codes of the four populations in all figures and tables. Response: We have now standardized names in Figures and Tables. We have also improved resolution of figures. Figures have been checked in PACE.

• Line 432-434: This hypothesis is not clear to me. I would suggest the authors explain it better. 

Response: We agree that this sentence might be confusing. We have now edited this hypothesis in the manuscript as follows:

“ A positive TD, could also reflects long-term events, where genetic traits that were selected during domestication might have been neutral or slightly deleterious in the wild ancestors. Because of this, these traits were present in the wild population due to a balance between random genetic changes and natural selection. This historical balance, combined with the selective pressure of domestication, can lead to an excess of intermediate alleles” (lines 434-439).

• Line 377-389: I would suggest the authors start the paragraph with "reliable conclusion and new ....even though we acknowledge ...", and then list the various cautions. Furthermore, I would suggest writing the paragraph more organically so that the various sentences do not seem like a list of answers to specific questions. 

Response: We have edited the introductory first paragraph of the Discussion (lines 378-391). 

Decision: accepted with minor revisions

Reviewer #2: The authors aimed to identify the effect of feralization in two cat populations inhabiting Dirk Hartog Island [Australia] and Kaho’olawe [Hawaii], respectively. The authors analysed the genome of 8 samples (4+4) and compared them with the available genome of domestic cats (n=11) widespread in Europe, America and Asia, and with wildcats (n=8) collected only from Germany, where the population suffered a strong decline.

Genomic analyses were correctly performed but I found critical issues in sampling, in some Results, and consequently, in the Discussion. I suggest reconsidering the analyses of more reference samples and avoiding analyses negatively influenced by the low sample size.

Moreover, the authors should consider softening some assumptions: e.g. the absolute absence of releasing of domestic cats after 1800. Particularly, Dirk Hartog Island is interested in ecotourism, so it can be hypothesized in the discussion the possibility that some results have been affected by rare and occasional cat releases. Higher caution should be used in interpreting the data from GWAS, as several variables could influence the results, and reference samples (both domestic and wildcats) are from very different habitats. It is hard to define which factors have been influencing adaptation events when different variables are acting differently in the studied groups. Some further comments:

We thank R#2 for carefully reading our manuscript and making useful comments and suggestions. We agree that this study would definitely benefit from a larger sample size, and it’s something that must be considered for future studies. We have acknowledged now in the manuscript the possibility that Dirk Hartog Island might not have been completely free of occasional and accidental cat releases after the initial introductions. 

We do not fully understand the reviewer’s concern about the data interpretation from GWAS. We do not perform a GWAS analysis in this study, we do not look at how specific genes are associated with particular traits. Although it would be very interesting for future studies, it’s not the purpose of the present work.

P5 L123 The authors wrote: “We included four feral cat samples per island”. As declared by the authors at the beginning of the Discussion, a sampling of only 8 feral cats can negatively influence the results. Response: We never claimed that our sampling can negatively influence the results. As mentioned above, we acknowledge our small sample size and that a larger sample size might provide more robust results. We would like nevertheless to highlight that we chose analyses that allow a low sample size and provided references on how low sample size does not necessarily affect the different analyses (in introductory paragraph of Discussion). In addition to the particular impact on the individual analysis, low sample size be justified by the use of a larger number of genetic markers used in our study. We draw caution on this issue in the introductory paragraph of the Discussion.

P5 L124 The authors wrote: “Both islands are not inhabited by humans and, domestic cats were introduced in the last 200 years”. This affirmation is not completely true as the Dirk Hartog Island [Australia] is known as an ecotourism destination. 

Response: We have now acknowledged the issue of ecotourism in Dirk Hartog Island in the Methods and provided a Reference which indicates no substantial introduction of cats in the near present (lines 141-145).

In addition, we want to be conservative with the initial assumption of a complete absence of cat releases (either accidental or purposedly) in Dirk Hartog Island. We addressed this issue in the Discussion in the context of inbreeding (lines 441-443), genetic diversity (line 425) and recombination rates (lines 480-482).

P5 L129 The authors wrote: “indicating similar environmental regimes”. There are several similarities in the ecological conditions of the two islands that authors recorded; however, Dirk Hartog Island [Australia] is quite three times larger than Kaho’olawe [Hawaii] and is human inhabited. 

Response: We have now addressed the larger size of Dirk Hartog in Methods (line 150) and in the Discussion, where we have previously not linked explicitly island size and the higher random drift in the smaller island Kaho’olawe (lines 551-553).

We comment on the issue of Dirk Hartog being human inhabited and ecotourism destination in the comment above.

P5 L130 The authors wrote: “included 8 German wildcat individuals”. Comparison with samples originating only in Germany can affect the bias. It must be considered that some features could be different in animals inhabiting island habitats and semi-arid climates. Differences in genetic variants could be recorded when analysing individuals inhabiting very different habitats. I suggest the introduction in the analysis of wildcats’ genomes from similar ecological areas; if it is not possible, I suggest adding adequate argumentations in the discussion of the results, particularly those referring to GWAS. 

Response: We agree with the reviewer that the inclusion in the study of wildcats from similar ecological areas might (or might not) have reduced different in some of the differences showed in this study between wildcats-domestic-feral cats. Unfortunately, only German WGS wildcats samples were available. 

We addressed reviewer’s GWAS concern above.

P6 L150 The authors wrote: “… and filtered for biallelic variants passing all filters”. I suggest adding information on applied filters, e.g. quality SNP and quality sample thresholds. 

Response: We have addressed and added additional information in this section (lines 169-170).

P6 L159 I suggest computing more K. 

Response: We have now repeated the ADMIXTURE analysis computing more Ks (1-5).

P11 L259 The authors should describe details about the genome data, e.g. reads, contigs, and so on. 

Response: We provide the GenBank accession number for the Felis catus genome used in this study in Methods (Line 167). We believe the genome statistics are easily accessible for readers seeking this information.

P11 L260 The authors should describe the total amount of SNP identified and how many were removed after using the different filter steps (e.g. for SNP or sample quality). 

Response: We have now added additional information of the initial number of SNPs identified (Line 270).

P 11 L262 The authors wrote: “two main groups”. It seems that the main groups are three. 

Response: We agree that this sentence was misleading. We referred to the first PC (PC1), that separates the wildcats and Felis catus individuals, which we discussed later. We have edited the start of the paragraph accordingly (line 273).

P11 L267-270 It is impossible to read the population names in the Admixture plot, so it is not possible to verify the genetic structure described in the text. The draft would benefit from the graphical view of the cv plot from 1 to 4K. CV values for K=3 should be reported. PCA describes 5 clusters, so I suggest computing a higher number of K and plotting the CV values to identify the best K. 

Response: We have replaced the ADMIXTURE plot according to the new analysis (which is now in a separate Figure as requested by Reviewer 3). We have also edited the ms and included a CV plot in Supplementary Information (Figure S1). We have amended CV error values in the manuscript as well. 

P11 L271-274 The authors wrote: “Hawaiian feral cats appeared sister to a clade of North American domestic cats”. In the supplementary material, SRRS35359631 is from Maui, so from Hawaii. There is a mistake in the tree because the sample from Iowa has been reported twice. Considering the position in the tree and the lack of the sample from Iraq, probably there was a switch of sample names. 

We thank the reviewer for spotting the mistake in one of the samples names in the tree. Indeed, the individual SRRS5359630 from Iowa was reported twice. The domestic cat individual from Iraq (SRRS5359639) is now in the tree.

P12 L275 Fst was computed between feral and domestic cats. Although wildcat sampling is only from Germany, it is important to carry out Fst computation also between feral and wildcats. As told before, the analysis would benefit from the inclusion of further wildcat samples. 

Response: As we have explained in Review#1, we believe that for future studies it would be more appropriate to do this comparison with F.lybica, the ancestor of F.catus, rather than with F. silvestris. In addition, we would like to point out that the main aim of the study is to look at different between feral vs domestic cats, rather than comparing feralization vs domes

---

## [Decision Letter · Decision Letter 1]

18 Jul 2024

PONE-D-24-04531R1Impact of feralization on evolutionary trajectories in the genomes of feral cat island populationsPLOS ONE

Dear Dr. Nieto Blázquez,

Thank you for submitting your manuscript to PLOS ONE. After careful consideration, we feel that it has merit but does not fully meet PLOS ONE’s publication criteria as it currently stands. Therefore, we invite you to submit a revised version of the manuscript that addresses the points raised during the review process.

We look forward to receiving your revised manuscript.

Kind regards,

Franck Courchamp

Academic Editor

PLOS ONE

Journal Requirements:

Additional Editor Comments:

The Authors clearly took into account all comments satisfactorily, and only a few minor editing points remain. Once these have been resolved, the article should be accepted without problem.

I thank the Authors for taking the review process seriously and producing this much improved revision. I trust this will be a significant contribution to the field.

Reviewers' comments:

Reviewer's Responses to Questions

**Comments to the Author**

1. If the authors have adequately addressed your comments raised in a previous round of review and you feel that this manuscript is now acceptable for publication, you may indicate that here to bypass the “Comments to the Author” section, enter your conflict of interest statement in the “Confidential to Editor” section, and submit your "Accept" recommendation.

Reviewer #1: All comments have been addressed

Reviewer #2: All comments have been addressed

Reviewer #3: All comments have been addressed

2. Is the manuscript technically sound, and do the data support the conclusions?

Reviewer #1: Yes

Reviewer #2: Yes

Reviewer #3: Yes

3. Has the statistical analysis been performed appropriately and rigorously? 

Reviewer #1: Yes

Reviewer #2: Yes

Reviewer #3: Yes

4. Have the authors made all data underlying the findings in their manuscript fully available?

Reviewer #1: Yes

Reviewer #2: Yes

Reviewer #3: Yes

5. Is the manuscript presented in an intelligible fashion and written in standard English?

Reviewer #1: Yes

Reviewer #2: Yes

Reviewer #3: Yes

6. Review Comments to the Author

Reviewer #1: (No Response)

Reviewer #2: The draft has been revised and improved. The main criticisms that have not been resolved have been adequately discussed. I have no further considerations but only the few listed below.

I suggest checking the punctuation and verifying the correctness of the symbols ( e.g. pi should be replaced with the correct symbol). The authors should verify to have introduced each symbol with the right definition before using it alone.

P17 L 404. I suggest the authors be careful in hypothesizing the same ship.

P17 L414 The authors wrote: "… the North American connection

of the feral cat population remains so far historically unexplained." I suggest including some hypotheses due to the origin of sailors or the simultaneous presence in European ports of ships coming from different routes.

P19 L451 The authors wrote: “Our results point to low levels of inbreeding in domestic cats, which challenges the general assumption that domesticated species show considerable inbreeding". This affirmation is true but it has to be considered that domestic cats can also derive from admixture between different breeds, justifying a higher variability.

Reviewer #3: From the revised manuscript, it is clear that the authors have taken my comments into account and made a concerted effort to correct the manuscript. I noticed only two small errors in the text:

- Line 437: Nucleotide diversity statistic (pi)... Please replace "pi" with the appropriate symbol for nucleotide diversity.

- Line 454: "wild cats" is separated again; please correct this.

With these minor corrections, the revised manuscript is acceptable for publication in the PLOS ONE journal.

7. PLOS authors have the option to publish the peer review history of their article (what does this mean?). If published, this will include your full peer review and any attached files.

Reviewer #1: No

Reviewer #2: No

Reviewer #3: No

---

## [Author Response · Author response to Decision Letter 1]

25 Jul 2024

Response to reviewers

Additional Editor Comments:

The Authors clearly took into account all comments satisfactorily, and only a few minor editing points remain. Once these have been resolved, the article should be accepted without problem.

I thank the Authors for taking the review process seriously and producing this much improved revision. I trust this will be a significant contribution to the field.

Response: We thank the Editor for acknowledging the authors effort to comply with the reviewers’ requirements. We are also grateful for their positive evaluation of our work.

We want to thank all 4 reviewers for their thorough and insightful comments on our manuscript.

Review Comments to the Author

Reviewer #1: (No Response)

Reviewer #2: The draft has been revised and improved. The main criticisms that have not been resolved have been adequately discussed. I have no further considerations but only the few listed below. Response: We appreciate the time and effort R2 has invested in providing valuable feedback, which has significantly improved the quality of our work.

I suggest checking the punctuation and verifying the correctness of the symbols (e.g. pi should be replaced with the correct symbol). The authors should verify to have introduced each symbol with the right definition before using it alone. Response: We have now checked the manuscript again for mistakes that we might have overlooked. We have now corrected mistakes in Line 183 (fixation index) and Line 383 (pi).

P17 L 404. I suggest the authors be careful in hypothesizing the same ship. Response: We agree with this comment. The sentence now reads as follows: “…it appears likely that the two feral cat populations stem from similar ship introductions with global routes spanning …” (Line 404).

P17 L414 The authors wrote: "… the North American connection

of the feral cat population remains so far historically unexplained." I suggest including some hypotheses due to the origin of sailors or the simultaneous presence in European ports of ships coming from different routes. Response: We acknowledge that providing well-formulated hypotheses for the observed phylogenetic pattern would be valuable. However, as this section is exploratory and not the primary focus of the paper, and due to the limited sample size, we prefer to avoid further speculation at this stage.

We agree that this is an interesting area for future research. We plan to address it more comprehensively in subsequent studies, where we can include a larger sample size and more historical information to conduct more in-depth phylogeographical analyses.

P19 L451 The authors wrote: “Our results point to low levels of inbreeding in domestic cats, which challenges the general assumption that domesticated species show considerable inbreeding". This affirmation is true but it has to be considered that domestic cats can also derive from admixture between different breeds, justifying a higher variability. Response: We thank the reviewer for this observation and have now incorporated this argument into the Discussion (Line 451-453).

Reviewer #3: From the revised manuscript, it is clear that the authors have taken my comments into account and made a concerted effort to correct the manuscript. I noticed only two small errors in the text:

Response: We appreciate the time and effort R3 has invested in providing valuable feedback, which has significantly improved the quality of our work.

- Line 437: Nucleotide diversity statistic (pi)... Please replace "pi" with the appropriate symbol for nucleotide diversity.

Response: We have now corrected this (Line 383).

- Line 454: "wild cats" is separated again; please correct this.

Response: We have now corrected this (Line 396 and 427).

---

## [Editor Report · Decision Letter 2]

30 Jul 2024

Impact of feralization on evolutionary trajectories in the genomes of feral cat island populations

PONE-D-24-04531R2

Dear Dr. Nieto Blázquez,

We’re pleased to inform you that your manuscript has been judged scientifically suitable for publication and will be formally accepted for publication once it meets all outstanding technical requirements.

Kind regards,

Franck Courchamp

Academic Editor

PLOS ONE
---

## [Editor Report · Acceptance letter]

2 Aug 2024

PONE-D-24-04531R2 

PLOS ONE

Dear Dr. Nieto-Blázquez, 

I'm pleased to inform you that your manuscript has been deemed suitable for publication in PLOS ONE. Congratulations! Your manuscript is now being handed over to our production team.

Kind regards, 

on behalf of

Dr. Franck Courchamp 

Academic Editor

PLOS ONE